# Molecular and Morphological Characterization of *Exserohilum turcicum* (Passerini) Leonard and Suggs Causing Northern Corn Leaf Blight of Maize in Bihar

**DOI:** 10.3390/bioengineering9080403

**Published:** 2022-08-19

**Authors:** Md Arshad Anwer, Ram Niwas, Tushar Ranjan, Shyam Sundar Mandal, Mohammad Ansar, Jitendra Nath Srivastava, Jitesh Kumar, Khushbu Jain, Neha Kumari, Aditya Bharti

**Affiliations:** 1Department of Plant Pathology, Bihar Agricultural University, Sabour 813210, Bhagalpur, India; ansar.pantversity@gmail.com (M.A.); j.n.srivastava1971@gmail.com (J.N.S.); neha.k1392@gmail.com (N.K.); adityabharti0806@gmail.com (A.B.); 2Department of Molecular Biology and Genetic Engineering, Bihar Agricultural University, Sabour 813210, Bhagalpur, India; mail2tusharranjan@gmail.com (T.R.); jitesh1jan@gmail.com (J.K.); khushbu3aug@gmail.com (K.J.); 3Department of Plant Breeding and Genetics, Bihar Agricultural University, Sabour 813210, Bhagalpur, India; maizebreederbau@gmail.com

**Keywords:** *β*-tubulin, *Setosphaeria turcica*, maize, turcicum leaf blight, Bihar

## Abstract

Maize is considered the third most important cereal crop in Asia after rice and wheat. Many diseases affect this crop due to the cultivation of various hybrids. This research aimed to characterize the causative agent of northern corn leaf blight disease in Bihar, India, caused by *Exserohilum turcicum* (Passerini) Leonard and Suggs. Leaf samples were collected from infected fields in five maize growing districts of Bihar in 2020–2022. A total of 45 fungal isolates from 135 samples were examined for cultural, morphological, and molecular characteristics and were identified as *E. turcicum*. The isolates were grouped into four groups based on colony color, i.e., olivaceous brown, blackish brown, whitish black, and grayish, and into two groups based on regular and irregular margins. The conidial shapes were observed to be elongated and spindle-shaped with protruding hilum, with conidial septa ranging from 2–12. Similarly, conidial length varied from 52.94 μm to 144.12 μm. *β*-tubulin gene sequences analysis made it possible to verify the identities of fungal strains and the phylogenetic relationships of all isolates, which were clustered in the same clade. The *β*-tubulin gene sequences of all the isolates showed a high level of similarity (100%) with reference isolates from GenBank accession numbers KU670342.1, KU670344.1, KU670343.1, KU670341.1, and KU670340.1. The findings of this study will serve as a baseline for future studies and will help to minimize yield losses.

## 1. Introduction

Maize is the third most significant cereal crop in India after rice and wheat. Numerous bacterial, viral, and fungal diseases can damage this crop. One of the significant foliar diseases is northern corn leaf blight (NCLB), which is caused by the fungus teleomorph—*Setosphaeria turcica* (Luttrell) and anamorph—*Exserohilum turcicum* (Passerini) Leonard and Suggs. The disease was initially described in Italy by Passerini in 1876 [1] and in 1907 by Butler in India [2]. Andhra Pradesh, Karnataka, Bihar, Himachal Pradesh, and Maharashtra are the states in India most affected by this disease. Early disease epidemics cause blighted leaves to die prematurely and lose their nutritional value, even as fodder. The majority of the composite and hybrid plants that are produced on a commercial scale are reported to be somewhat vulnerable to NCLB. In cooler maize-growing locations, NCLB is an endemic disease that is regarded to be crucial in terms of its geographic prevalence and ability to reduce production. When the leaves over the ear are impacted even very slightly during the post-flowering period, losses from NCLB are more severe. The most severe disease impacts are produced by a warm, humid climate; late planting; and maize grown from previous seasons [3]. NCLB is among the most devastating foliar diseases, causing serious diminishment in grain yield of around 16–98% [4]. In mid-elevation tropical zones with low temperatures, cloudy weather, and excessive humidity during the maize growing season, NCLB can be a major problem [5]. If the symptoms appear prior to flowering, yield losses may exceed 50% [6,7]. The infection manifests as boat-shaped blighting and long gray, green, or brown elliptical streaks. The primary determinant of host-plant resistance and the development of effective disease management strategies are the genetic variability and pathogenicity of the pathogen. This pathogen’s virulence has been reported to vary in maize [8] and, more recently, in sorghum [9]. In sorghum, three RAPD markers that are tightly linked to a locus for resistance to NCLB have been discovered [10]. Little is known about the variations in isolates of *E. turcicum* at the molecular level [11]. However, according to Mathur [12], there are no such reports on the existence of various *E. turcicum* races in sorghum. In order to standardize molecular markers helpful for such research and ascertain the degree of genetic variability in this disease, an assessment was required. In studying the ecology and biology of fungi, RAPD and SSR markers are useful for determining genetic similarity and identifying variation within and among populations of *E. turcicum* [11,13] and other fungal species [14,15]. Numerous researchers have tried to identify the specific pathogen that causes NCLB disease. However, in this study, cultural and morphological variability of *E. turcicum* among different isolates causing NCLB disease in maize was identified and isolates’ molecular characterization was done in order to better and more easily identify this pathogen via morphological and molecular diversity, as well as to emphasize the development of disease-resistant cultivars.

## 2. Materials and Methods

### 2.1. Sample Collection and Isolation of Pathogens under Standard Cultivation Conditions

In the present study, *E. turcicum* isolates were collected in 2020–2022 from five districts of Bihar; three blocks from each district and nine villages from each block were chosen for sample collection. The details of the districts surveyed and the areas from which diseased isolates were collected for variability studies are given in Table 1. Field samples were labeled, kept in cold boxes, and brought to the laboratory for further study [16].

### 2.2. Isolation and Identification of Exserohilum Turcicum

The isolation of fungus from diseased samples was carried out using the method described by Manamgoda [17]. Small portions of infected leaf tissue with some adjacent healthy tissues of around 0.5 cm × 0.5 cm in diameter were cut. The leaf portions were surface sterilized with 1% sodium hypochlorite (NaOCl) for 30 s. The desired portions were removed with sterilized forceps and transferred into distilled water for 1–2 min. The portions were blotted on sterilized filter paper in order to absorb moisture, and finally the portions were placed on a suitable nutrient potato dextrose agar (PDA) medium and the plates were completely sealed with Para film^®^, followed by incubation at 26 ± 2 °C. Colonies were observed at 2–3 day intervals until full growth was attained. Small portions of fungal mycelium from fully grown culture were aseptically transferred to plates of fresh PDA culture medium in order to obtain pure cultures of *E. turcicum* [16,18].

### 2.3. Cultural and Morphological Characterization

Growing cultures of 7–10 days were used for this study. The growing cultures in Petri plates were aseptically opened under laminar air flow, clean microscope glass slides were gently placed over the surface of the colonies, touching the edge, and the plates were resealed, followed by incubation under continuous light to produce spores. A total of 45 fungal isolates were studied for morphological and cultural characteristics after the culture growth reached 10 days [18,19]. The color and texture of the colonies as well as growth pattern, pigmentation, and margin growth were observed [19]. In the conidial study, number of septa, conidial length, color, and width of 30 spores per isolate were measured using a fluorescent microscope (Evos, Thermo Fisher Scientific, Waltham, MA, USA) and analyzed using Image J software [20].

### 2.4. Molecular Characterization

#### 2.4.1. DNA Isolation

Conical flasks with 2% potato dextrose broth were used to culture *E. turcicum* isolates for 10 days at 26 ± 2 °C. A few changes to the modified CTAB procedure [21,22] were made in order to extract DNA from the growing mycelial mats. Using liquid nitrogen, the collected fungus mats were ground into a fine powder. The extracts were transferred to sterile polypropylene tubes with 15 mL of 2× CTAB extraction buffer, which contains 2–3 percent *W/V* CTAB, 1.4 M NaCl, 20 mM EDTA, 100 mM Tris-HCl, pH-8, and 0.17% beta-mercaptoethanol, preheated to 65 °C. The DNA was purified using equal volumes of ethanol and chloroform-isoamyl alcohol. Centrifugation was then done at 10,000 rpm for 5 min and the supernatant was transferred to a new tube. Genomic DNA was precipitated by adding ice-chilled iso-propenol (0.6 mL), mixed by inversion, and incubated in ice for 30 min. The samples were then centrifuged at 10,000 rpm for 10 min at 4 °C. Supernatant was discarded and DNA was washed with 70% ethanol. The pellets were air dried and re-suspended in 100 µL of TE buffer [21,23].

#### 2.4.2. PCR Amplification

The modified CTAB protocol was followed in order to extract the DNA, and then PCR was used to amplify the DNA using the universal primers TUBUF2 forward (5′-CGGTAACAACTGGGCCAAGG-3′) and TUBUR1 reverse (5′-CCTGGTACTGCT GGTACTCAG-3′) [24]. A thermal cycler was used to carry out the PCR amplification. The overall volume for the PCR reaction was 30 μL, which included 2 μL of DNA template, 1.5 μL of forward and reverse primer [24], 10 μL of nuclease-free water, and 15 μL of PCR master mix (Taq polymerase). The thermo cycler (Gradient thermal cycler Master cycler^®^ nexus) consisted of an initial denaturation step at 95 °C for 4 min, 30 cycles of denaturation at 95 °C for 30 s, annealing at 56.6 °C for 30 s, extension at 72 °C for 1 min, and final extension at 72 °C for 5 min. Gel electrophoresis and staining were carried out by adding 10 μL of PCR product and 1% of agarose gel to a TAE buffer solution (40 mM Tris, 20 mM acetic acid, and 1 mM EDTA) and running the mixture at 80 V for 50 min at 25 °C. The gel was stained with Fluorosafe stain and the PCR products were observed under a UV light and documented using a gel documentation system (Invitrogen, Thermo Fisher, Scientific, Waltham, MA, USA). A molecular marker (DNA ladder mix, 1 kb, Genbiotech, SRL) was used to determine the size of amplified DNA bands. The PCR products were finally sent for sequencing, which was done by Sanger sequencing method (Illumina, San Digo, CA, USA) [25]. Multiple sequence alignments were generated using Clustal Omega (http://www.ebi.ac.uk/Tools/msa/clustalo/) accessed on 11 August 2022 [26] and were also manually corrected for domain superimpositions. MEGA11 software was used to align the *β*-tubulin sequences, and BLASTN search was used to compare them to sequences in the GenBank database (http://www.ncbi.nlm.nih.gov) accessed on 11 August 2022 [27]. All of the isolate accession numbers were taken from GenBank. MEGA11 software was used to create a phylogenetic tree [28].

## 3. Results

### 3.1. Cultural and Morphological Characterization

The colony colors varied from olivaceous brown to a whitish black color. Based on the colony color, all 45 isolates were grouped into four categories, i.e., (i) olivaceous brown (BhEt2, BhEt4, BeEt2, BeEt4, BeEt5, BeEt8, KhEt2, KhEt5, KhEt7, KhEt8, KaEt1, KaEt3, KaEt8, KaEt9, SaEt3, and SaEt9), (ii) blackish brown (BhEt1, BeEt3, SaEt4, and SaEt5), (iii) whitish black (BhEt3, BhEt5, BhEt6, BhEt7, BhEt8, BeEt1, KhEt6, KaEt2, and KaEt5), and (iv) grayish (BhEt9, BeEt6, BeEt7, BeEt9, KhEt1, KhEt3, KhEt4, KhEt9, KaEt4, KaEt6, KaEt7, SaEt1, SaEt2, SaEt6, SaEt7, and SaEt8) (Figure 1 and Figure 2, Table 1).

Based on the colony type, all 45 isolates were grouped into four categories, i.e., (i) flattened (BhEt1, BhEt2, BhEt4, BeEt2, BeEt6, BeEt7, KhEt1, KhEt3, KhEt7, KhEt8, KhEt9, SaEt3, SaEt5, SaEt8, and SaEt9), (ii) raised cottony (BhEt5, BhEt6, BhEt7, and BhEt8), (iii) slightly raised fluffy (BeEt1, BeEt3, BeEt4, BeEt5, BeEt8, BeEt9, KhEt5, KaEt1, KaEt2, KaEt9, and SaEt4), and (iv) fluffy raised cottony (BhEt3, BhEt9, KhEt2, KhEt4, KhEt6, KaEt3, KaEt4, KaEt5, KaEt6, KaEt7, KaEt8, SaEt1, SaEt2, SaEt6, and SaEt7). 

The highest (90.0 mm) growth was observed in the BhEt1, BeEt3, BeEt6, BeEt7, BeEt9, KhEt1, KhEt2, KhEt3, KhEt8, KhEt9, KaEt4, KaEt6, KaEt8, SaEt1, SaEt2, SaEt5, SaEt6, and SaEt7 isolates followed by the SaEt4 isolate (89.9 mm), whereas the lowest (64.6 mm) growth was observed in BeEt5 followed by the isolate KaEt5 (64.7 mm) on 10 days after inoculation (Figure 1 and Figure 2, Table 1). 

Based on colony margin, all the *E. turcicum* isolates were classified into two groups, i.e., regular margin and irregular margin. The isolates BhEt1, BhEt2, BhEt3, BhEt4, BhEt5, BhEt6, BhEt7, BhEt8, BeEt2, BeEt3, BeEt4, BeEt6, BeEt7, BeEt9, KhEt1, KhEt2, KhEt3, KhEt4, KhEt7, KhEt8, KhEt9, KaEt1, KaEt3, KaEt4, KaEt5, KaEt6, KaEt7, KaEt8, KaEt9, SaEt1, SaEt2, SaEt3, SaEt4, SaEt5, SaEt6, SaEt7, SaEt8, and SaEt9 came under the group of colonies with regular margins, while irregular margins were observed in the isolates BhEt9, BeEt1, BeEt5, BeEt8, KhEt5, KhEt6, and KaEt2 (Figure 1 and Figure 2, Table 1).

Based on margin color, all the *E. turcicum* isolates were categorized into four groups, i.e., gray, brown, white, and black color. The isolates BhEt9, KhEt1, KhEt3, KhEt4, KhEt7, KhEt9, KaEt4, KaEt6, KaEt7, KaEt8, SaEt1, SaEt2, SaEt6, SaEt7, and SaEt8 came under the group of colonies with a gray color margin; isolates BhEt1, BhEt2, BhEt4, BeEt2, BeEt3, BeEt4, KhEt5, KhEt8, KaEt9, SaEt3, SaEt4, SaEt5, and SaEt9 came under the group of colonies with a brown color margin; isolates BhEt3, BhEt5, BhEt6, BhEt7, BhEt8, BeEt5, KhEt2, KaEt1, KaEt3, and KaEt5 came under the group of colonies with a white color margin; and isolates BeEt1, BeEt6, BeEt7, BeEt8, BeEt9, KhEt6, and KaEt2 came under the group of colonies with a black color margin (Figure 1 and Figure 2, Table 1).

Based on pigmentation, *E. turcicum* isolates were grouped into three groups, i.e., (i) brownish (BhEt1, BhEt2, BhEt4, BeEt2, BeEt4, BeEt9, KhEt2, KhEt5, KhEt8, KaEt1, KaEt3, KaEt9, SaEt3, and SaEt9), (ii) grayish black (BhEt3, BhEt9, BeEt3, BeEt6, BeEt7, KhEt1, KhEt3, KhEt4, KhEt9, KaEt4, KaEt6, KaEt7, SaEt1, SaEt2, SaEt4, SaEt5, SaEt6, SaEt7, and SaEt8), and (iii) whitish black (BhEt5, BhEt6, BhEt7, BhEt8, BeEt1, BeEt5, BeEt8, KhEt6, KhEt7, KaEt2, KaEt5, and KaEt8) (Figure 1 and Figure 2, Table 1).

The conidial septa varied from 2–12, with the maximum number observed for isolates KhEt1, KhEt5, KaEt4, SaEt2, and SaEt3 (3–12), followed by BeEt7 (3–11), BeEt9, KhEt4, KhEt6, KhEt8 KaEt6, SaEt4, SaEt7, and SaEt9 (3–10), and the minimum in BhEt9, BeEt2, and KaEt1 (2–5) followed by KhEt9 (2–6). Most isolates (approximately 20) with conidial septa ranged from 3 to 6, 3 to 7, 3 to 8, or 3 to 9 (Figure 3, Table 1). 

Conidial length also varied from 52.94–144.12 µm and conidial width from 10.0–25.88 µm. The largest conidial length and width was recorded in SaEt4 (144.12 × 25.88 µm) followed by BhEt6 (137.65 × 16.47 µm and KhEt2 (130.5 × 24.12 µm), and the smallest conidial length was recorded in SaEt9 (52.94 × 11.18 µm) followed by SaEt1 (55.29 × 15.29 µm) and BhEt3 (55.88 × 11.18 µm) (Figure 3, Table 1).

Based on conidial shape, all the *E. turcicum* isolates were classified into two groups, i.e., elongated, ellipsoidal, obclavate to fusiform, slightly curved, spindle-shaped with protruding hilum (BhEt1, BhEt2, BhEt4, BhEt5, BhEt6, BeEt1, BeEt2, BeEt3, BeEt5, BeEt6, BeEt7, BeEt8, BeEt9, KhEt1, KhEt2, KhEt3, KhEt4, KhEt5, KhEt6, KhEt7, KhEt8, KhEt9, KaEt1, KaEt2, KaEt3, SaEt3, SaEt4, SaEt6, SaEt7, SaEt8, and SaEt9) or elongated, ellipsoidal, obclavate to fusiform, spindle-shaped with protruding hilum (BhEt3, BhEt7, BhEt8, BhEt9, BeEt4, KaEt4, KaEt5, KaEt6, KaEt7, KaEt8, KaEt9, SaEt1, SaEt2, and SaEt5) (Figure 3, Table 1).

Based on the conidial color, all the *E. turcicum* isolates were classified into three groups, i.e., brown, dark brown, and olivaceous brown. The isolates BhEt1, BhEt4, BhEt6, BeEt6, KhEt1, KhEt5, KhEt6, KhEt9, KaEt1, KaEt3, KaEt4, KaEt5, SaEt3, SaEt4, SaEt8, and SaEt9 came under the group of conidia with brown color; the isolates BhEt2, BhEt3, BhEt7, BhEt8, BhEt9, BeEt2, BeEt4, BeEt7, KhEt2, KhEt3, KhEt4, KhEt7, KaEt6, SaEt1, SaEt2, SaEt5, and SaEt7 came under the group of conidia with dark brown color; and the isolates BhEt5, BeEt1, BeEt3, BeEt5, BeEt8, BeEt9, KhEt8, KaEt2, KaEt7, KaEt8, KaEt9, and SaEt6 came under the group of conidia with olivaceous brown color (Figure 3, Table 1).

The number of conidia/microscopic field varied from 8–26. The maximum number of conidia was recorded in KhEt5 and SaEt8 (26.0), followed by KaEt4 (25.0), whereas the minimum number of conidia was recorded in BhEt7, BeEt1, BeEt4, BeEt9, KaEt1, and KaEt2 (8.0), followed by BhEt4, BeEt8, and KhEt3 (9.0) (Figure 3, Table 1).

The number of conidia/mL varied from 3.08–9.88 × 10^5^. The maximum number of conidia was recorded in BeEt6 and SaEt1 (9.88 × 10^5^), followed by SaEt2 (9.84 × 10^5^) and KhEt8 (9.78), whereas the minimum number of conidia was recorded in KaEt5 (3.08 × 10^5^), followed by KaEt2 (3.12 × 10^5^) (Figure 3, Table 1).

On the basis of morphological and cultural characterization, all the *E. turcicum* isolates were classified into six clades using their similarity coefficients (Figure 4). The isolates BhEt4, BhEt5, BhEt6, BhEt7, BhEt8, BhEt9, KhEt1, KhEt2, KhEt3, KhEt4, KhEt5, KaEt1, and KaEt3 came under clade I; the isolates BeEt5, BeEt6, BeEt8, BeEt9 SaEt6, SaEt7, and SaEt9 came under clade II; the isolates KhEt6, KhEt9, KaEt2, KaEt5, SaEt1, SaEt2, and SaEt3 came under clade III; the isolates BeEt1, BeEt2, BeEt3, BeEt7, KhEt7, KhEt8, KaEt4, KaEt6, KaEt7, KaEt8, KaEt9, and SaEt5 came under clade IV; the isolates BhEt3 and SaEt8 came under clade V; and the isolates BhEt1, BhEt2, BeEt4, and SaEt4 came under clade VI (Figure 4).

### 3.2. Molecular Characterization

Results from molecular identification indicated that 45 of the isolates were identified as *E. turcicum* (*Setosphaeria turcica*). Representative bands of PCR products were viewed using a gel documentation system (Figure 5). *β*-tubulin gene sequences analysis makes it possible to verify the identities of fungal strains. However, up to now there have been limited numbers of *β*-tubulin rDNA sequences in the public databases. In this study, it was therefore not possible to resolve the taxonomic affiliation. Figure 6 represents the phylogenetic relationships of all 45 isolates; all the isolates were clustered in the same clade. The *β*-tubulin gene sequences of BhEt1, BhEt2, BhEt3, BhEt4, BhEt5, BhEt6, BhEt7, BhEt8, BhEt9, BeEt1, BeEt2, BeEt3, BeEt4, BeEt5, BeEt6, BeEt7, BeEt8, BeEt9, KhEt1, KhEt2, KhEt3, KhEt4, KhEt5, KhEt6, KhEt7, KhEt8, KhEt9, KaEt1, KaEt2, KaEt3, KaEt4, KaEt5, KaEt6, KaEt7, KaEt8, KaEt9, SaEt1, SaEt2, SaEt3, SaEt4, SaEt5, SaEt6, SaEt7, SaEt8, and SaEt9 showed a high level of similarity (100% identity) with the reference isolates from GenBank (accession number KU670342.1, KU670344.1, KU670343.1, KU670341.1, KU670340.1). 

### 3.3. Sequence Analysis and Nucleotide Data Submission in GenBank

The sequence data were assembled and analyzed using Mega11 software. A sequence similarity Basic local alignment search tool (BLAST) was performed by comparing the sequence to other *E. turcicum* sequences using the tool’s website (http://www.ncbi.nlm.nih.gov/BLAST/) accessed on 11 August 2022. The data were subjected to multiple sequence alignments and a phylogenetic tree was constructed using Clustal Omega (http://www.ebi.ac.uk/Tools/msa/clustalo/) accessed on 11 August 2022 and Mega11 software. The sequence data of *E. turcicum* were submitted to the GenBank under the accession numbers ON813160 (SaEt9), ON813161 (SaEt8), ON813162 (SaEt7), ON813163 (SaEt6), ON813164 (SaEt5), ON813165 (SaEt4), ON813166 (SaEt3), ON813167 (SaEt2), ON813168 (SaEt1), ON813169 (KaEt9), ON813170 (KaEt8), ON813171 (KaEt7), ON813172 (KaEt6), ON813173 (KaEt5), ON813174 (KaEt4), ON813175 (KaEt3), ON813176 (KaEt2), ON813177 (KaEt1), ON813178 (KhEt9), ON813179 (KhEt8), ON813180 (KhEt7), ON813181 (KhEt6), ON813182 (KhEt5), ON813183 (KhEt4), ON813184 (KhEt3), ON813185 (KhEt2), ON813186 (KhEt1), ON813187 (BeEt9), ON813188 (BeEt8), ON813189 (BeEt7), ON813190 (BeEt6), ON813191 (BeEt5), ON813192 (BeEt4), ON813193 (BeEt3), ON813194 (BeEt2), ON813195 (BeEt1), ON813196 (BhEt9), ON813197 (BhEt8), ON813198 (BhEt7), ON813199 (BhEt6), ON813200 (BhEt5), ON813201 (BhEt4), ON813202 (BhEt3), ON813203 (BhEt2), and ON813204 (BhEt1).

Multiple sequence alignment of all 45 *E. turcicum* isolates indicated the presence of highly conserved regions (highlighted in black) throughout the *β* tubulin gene (Figure 7). Intriguingly, variable regions were also found in these isolates (highlighted in red). This suggests the divergence of different isolates of *E. turcicum* during the course of its evolution due to insertion or deletion events. The complete alignment file along with the details of consensus and variable regions is also depicted in Appendix A.

## 4. Discussion

All 45 isolates of *E. turcicum* did differ in various aspects such as incubation period in days for maximum growth, colony color, pigmentation, sporulation, colony texture, color of conidia, number of conidia/microscopic field, and number of conidia/mL. Gowda reported similar variations [29,30]. Conidia measured 52.94–144.12 µm; were fusoid, smooth, straight, or curved; generally 2–12 septate; and had a small, protruding basal hilum. These observations of morphological variation among various isolates are consistent with those made by Misra and his coworker [31,32,33,34,35]. Despite these changes, the conidial size (50–144 µm) and septa of 4–9 of *E. turcicum* were within the usual described range of conidial size [36]. The morphological variations between the isolates of *E. turcicum* (*Setosphaeria turcica*) isolated from maize have been compared by many researchers [33,37,38,39,40]. In this study, conidia size, radial growth, and pigmentation are the PCA factors that contributed highest for clustering. It was noted that the conidial forms were elongated and spindle-shaped. Cultural traits demonstrated that there were differences in colony development and color among the isolates. Based on the color of the colony, the isolates were divided into three groups: light gray, gray, and dark gray. Based on growth, the isolates were categorized into two groups: those with moderate growth and those with profuse growth. The isolates ET002 and ET003 presented septa with a range of 5–7 to 7–10, respectively. Similarly, isolates ET002 and ET003 had conidial lengths that ranged from 56.7 m to 89.44 µm, related to this study [41]. The variability among isolates of *E. turcicum* may be due to ability of the pathogen to adapt, with variations developed in particular conditions and due to the long-term influence of the weather in a particular location [42]. Thus, it clearly indicates the existence of different strains/virulence within *E. turcicum* [43]. Thus, it has been made abundantly evident that there are many strains and levels of variation within *E. turcicum* [44,45,46]. In general, recombination and mutations are the primary causes of fungal genetic variation [47,48]. Low mutation rates result from the rarity of avirulence to virulence mutations [49]. The variation of the *E. turcicum* population in the tropics may be the result of sexual recombination as well [50]. Somatic recombination, particularly in temperate locations, may also be a source of genetic variation in *E. turcicum* populations [47]. Another ascomycete, *Magnaporthe grisea*, which infects grasses and causes blast disease in rice, has been characterized in the literature as having parasexual characteristics [51]. However, more research is required to establish the existence of parasexuality in *E. turcicum* and to determine how mixed reproduction contributes to the race variability of this pathogen. 

Results from molecular identification indicate that 45 isolates were identified as *Setosphaeria turcica*-telomorph (*E. turcicum*—anamorph). A gel documentation system was used to observe representative PCR product bands. Analysis of the *β*-tubulin gene sequences allowed the identification of different fungal strains. In this study, the *β*-tubulin genes of all 45 isolates were closely related to the *β*-tubulin gene sequences of reference isolates from Gen Bank (accession numbers KU670342.1, KU670344.1, KU670343.1, KU670341.1, KU670340.1). Similar findings were reported for *Alternaria infectoria*, which acted as an outgroup in this study because it belonged to neither of the two clades [41]. Phylogenetic analysis and DNA-based molecular methods have changed how this fungal group is classified, among other things. For example, molecular analysis employing the internal transcribed spacer (ITS) region of nuclear ribosomal DNA has augmented the conventional means of categorization by offering an accurate and quick means of species identification from distinctive hosts [52,53]. Burdon and Silk suggest that mutation and recombination are the primary sources of genetic variation in plant pathogenic fungi [54]. When studying the ecology and biology of fungi, RAPD and SSR markers are useful for determining genetic similarity and identifying variation within and among populations of *E. turcicum* [11,13]. These mechanisms are supplemented within a species by gene flow across populations as propagules move from one region to another. Gene flow and other evolutionary factors can cause the spread of certain genes or DNA sequences as well as the formation of entire populations in various geographical areas. This variation is supported by pressures of selection and genetic drift, further increasing the evolutionary divergence [55]. In their initial study of the sexual stage of *E. turcicum* in Thailand, Bunkoed hypothesized that sexual reproduction in *Setosphaeria turcica* had contributed to genetic variation in the fungal pathogen [56]. This hypothesis was confirmed by earlier research using inter simple sequence repeat markers. Due to host–pathogen interactions and environmental constraints, the cultivation of several maize types may have resulted in greater selection pressure, which in turn improved diversity in the pathogen population structure. 

## 5. Conclusions

At the end of this study, the variability of the *E. turcicum* pathogen responsible for causing NCLB disease in Bihar was studied; the pathogen was identified as *E. turcicum* using both morphological and molecular methods. *E. turcicum* was found to be capable of growing in a wide range of agroclimatic areas of Bihar. Moreover, the 45 isolates were categorized into six groups on the basis of cultural and morphological variability. This pathogen also showed molecular variability among the species of *Setosphaeria turcica*. However, variation in the *β*-tubulin gene sequence among the isolates did not show any relationship with the morphological and cultural variations. Therefore, this study will help plant pathologists in the future to understand more about the variability of this pathogen, and might be helpful in understanding this fungus which could help in its control. The present study also compiled both morphological and molecular data of *E. turcicum* together for the first time. The study will also serve as the baseline for in vitro and in vivo research into NCLB disease of maize in the future.

## Figures and Tables

**Figure 1 bioengineering-09-00403-f001:**
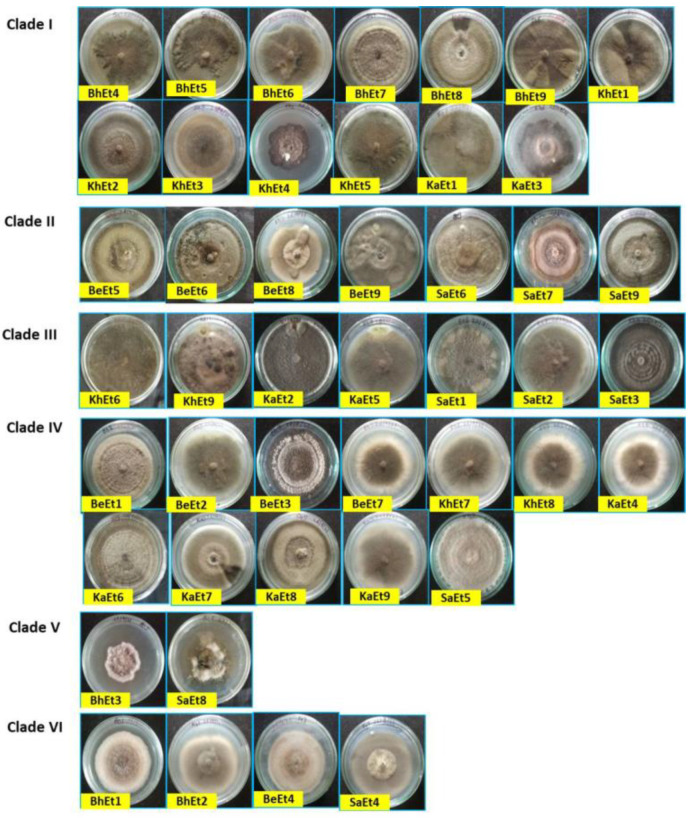
Cultural and morphological variability of different isolates of *E. turcicum* on PDA (front view) 10 days after inoculation.

**Figure 2 bioengineering-09-00403-f002:**
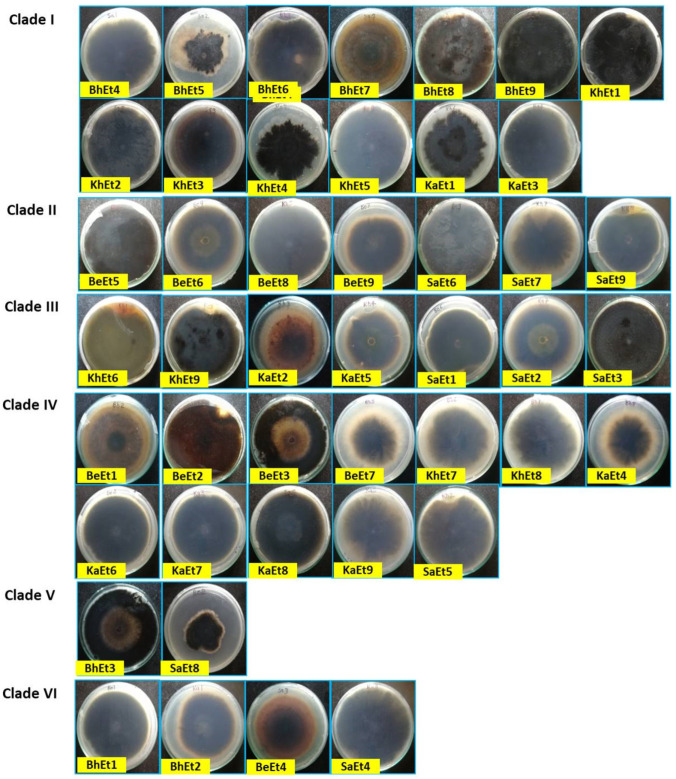
Cultural and morphological variability of different isolates of *E. turcicum* on PDA (inverted view) 10 days after inoculation.

**Figure 3 bioengineering-09-00403-f003:**
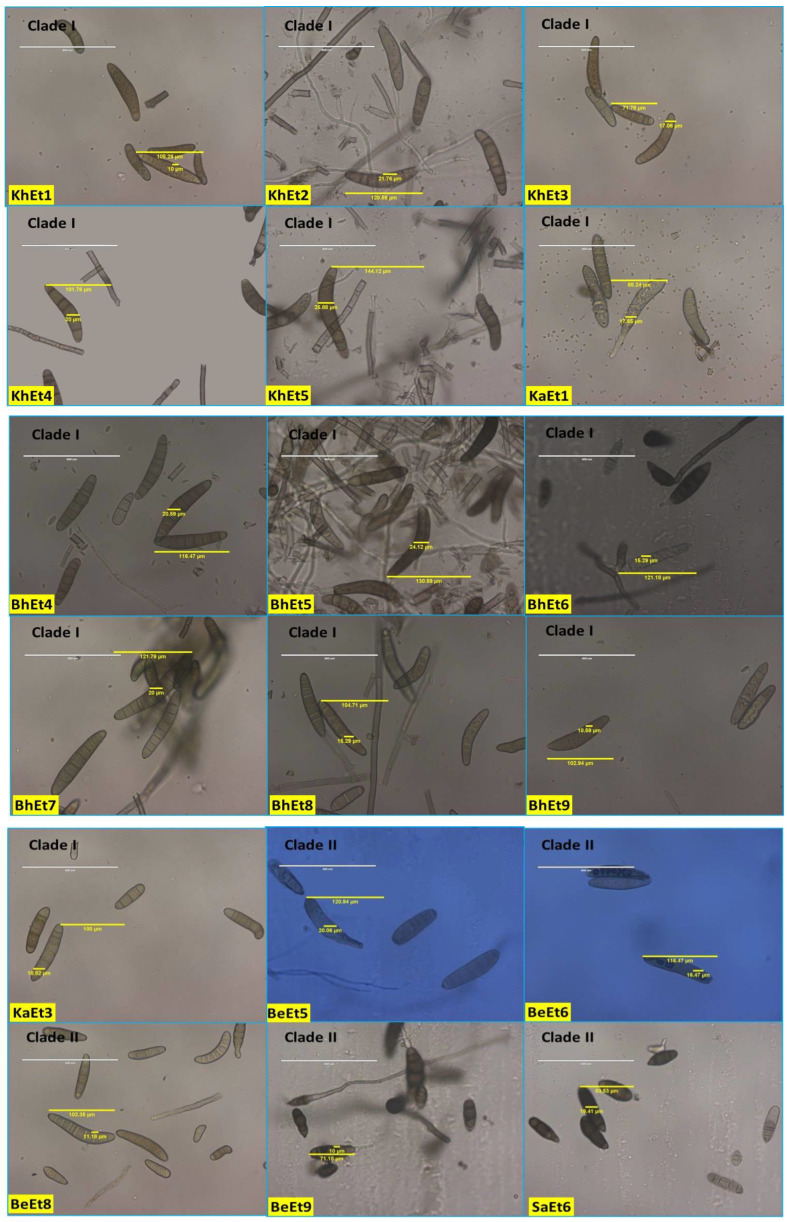
Conidial variability of different isolates of *E. turcicum* causing NCLB.

**Figure 4 bioengineering-09-00403-f004:**
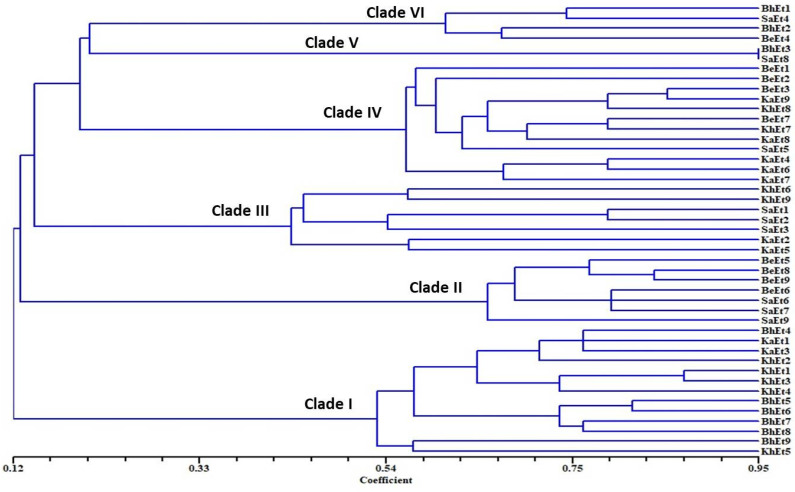
Dendrogram on the basis of morphological and cultural variability of 45 *E. turcicum* isolates causing NCLB disease.

**Figure 5 bioengineering-09-00403-f005:**
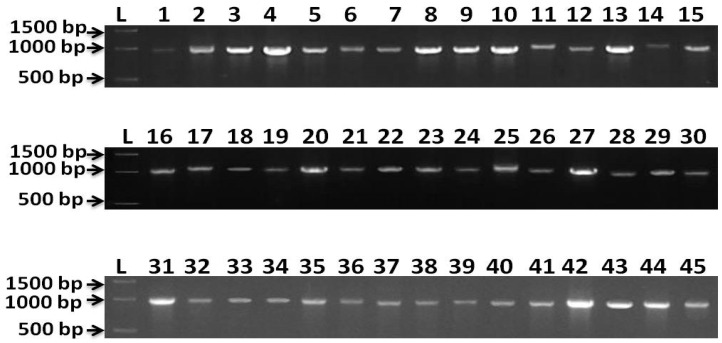
Representative bands of PCR product from β-tubulin gene. Amplification fragment size 1000 base pairs (bp). L = 1kb ladder, 1 = BhEt1, 2 = BhEt2, 3 = BhEt3, 4 = BhEt4, 5 = BhEt5, 6 = BhEt6, 7 = BhEt7, 8 = BhEt8, 9 = BhEt9, 10 = BeEt1, 11 = BeEt2, 12 = BeEt3, 13 = BeEt4, 14 = BeEt5, 15 = BeEt6, 16 = BeEt7, 17 = BeEt8, 18 = BeEt9, 19 = KhEt1, 20 = KhEt2, 21 = KhEt3, 22 = KhEt4, 23 = KhEt5, 24 = KhEt6, 25 = KhEt7, 26 = KhEt8, 27 = KhEt9, 28 = KaEt1, 29 = KaEt2, 30 = KaEt3, 31 = KaEt4, 32 = KaEt5, 33 = KaEt6, 34 = KaEt7, 35 = KaEt8, 36 = KaEt9, 37 = SaEt1, 38 = SaEt2, 39 = SaEt3, 40 = SaEt4, 41 = SaEt5, 42 = SaEt6, 43 = SaEt7, 44 = SaEt8, 45 = SaEt9.

**Figure 6 bioengineering-09-00403-f006:**
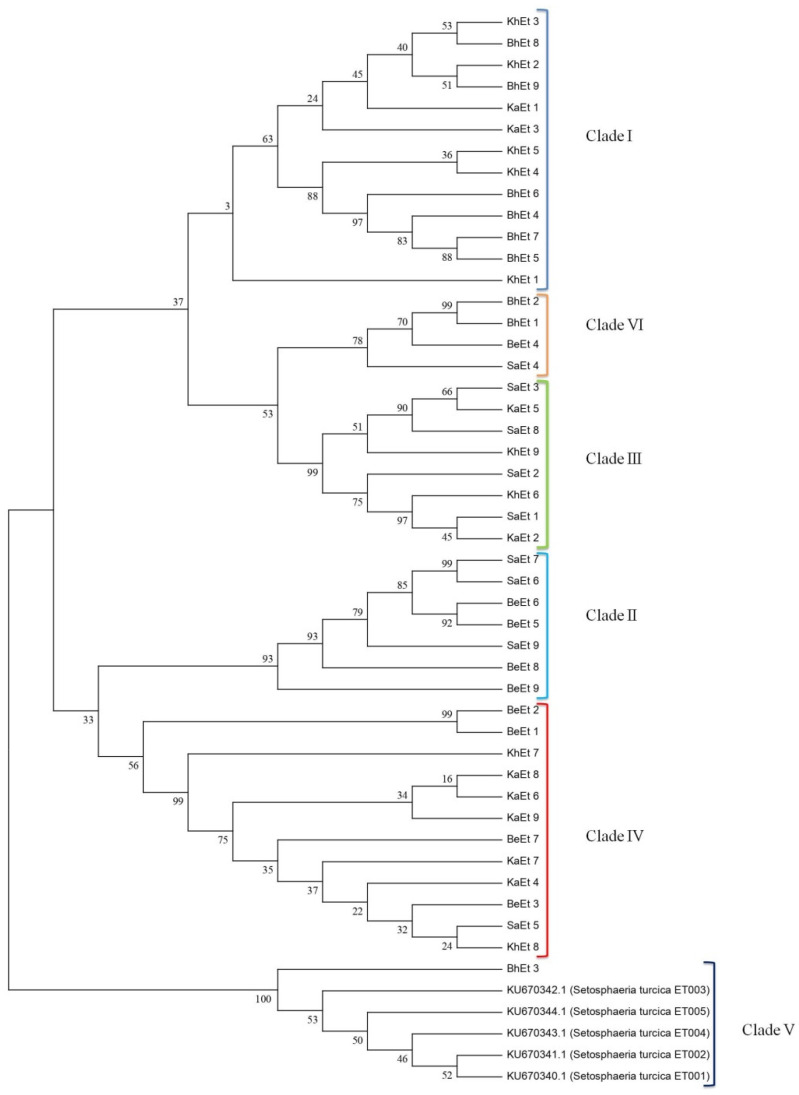
Phylogenetic relationships of 45 *E. turcicum* isolates compared with accession numbers of other species. Phylogenetic tree inferred by maximum likelihood analysis based on rDNA sequences. Numbers below the branches represent the percentage for each branch in 1000 bootstrap replications.

**Figure 7 bioengineering-09-00403-f007:**
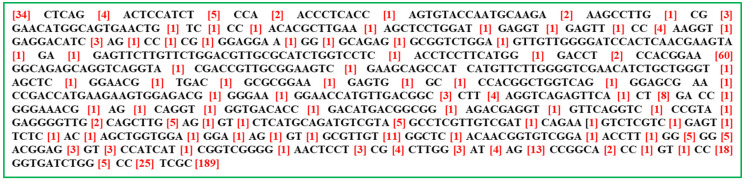
Consensus sequence (highlighted in black) present in the *β*-tubulin gene of all 45 *E. turcicum* isolates with variable regions in brackets (highlighted in red), fetched after multiple sequence alignment using Clustal Omega. Details of alignment are also given in Appendix A.

**Table 1 bioengineering-09-00403-t001:** Cultural and morphological variability among 45 isolates of *E. turcicum* causing NCLB disease of maize in Bihar.

Isolates	Colony						Conidia					Sporulation /Microscopic Field	Conidia/mL
	Type	Color	Radial Growth (mm)	Margin	Pigmentation	Margin Color	Color	No. of Septa	Length (µm)	Width (µm)	Shape		
**BhEt1**	Flattened	Blackish brown	90.0	Regular	Brownish	Brown	Brown	3–6	101.76	20	Elongated, ellipsoidal, slightly curved fusiform with protruding hilum	12	9.75 × 10^5^
**BhEt2**	Flattened	Olivaceous brown	88.1	Regular	Brownish	Brown	Dark Brown	3–9	72.35	15.29	Slightly curved	18	9.22 × 10^5^
**BhEt3**	Fluffy Raised cottony	Whitish black	67.3	Regular	Grayish Black		Dark Brown	3–8	55.88	11.18	Elongated, ellipsoidal, obclavate to fusiform, spindle with protruding hilum	22	5.45 × 10^5^
**BhEt4**	Flattened	Olivaceous brown	85.4	Regular	Brownish	Brown	Brown	3–7	71.76	17.06	Slightly curved	9	8.12 × 10^5^
**BhEt5**	Raised Cottony	Whitish black	70.3	Regular	Whitish Black	White	Olivaceous brown	3–10	120	13.53	Slightly curved	15	5.55 × 10^5^
**BhEt6**	Raised Cottony	Whitish black	75.6	Regular	Whitish Black	White	Brown	3–7	137.65	16.47	Slightly curved	12	6.8 × 10^5^
**BhEt7**	Raised Cottony	Whitish black	85.8	Regular	Whitish Black	White	Dark Brown	3–7	121.18	15.29	Obclavate	8	8.65 × 10^5^
**BhEt8**	Raised Cottony	Whitish black	77.5	Regular	Whitish Black	White	Dark Brown	3–8	117.65	21.18	Obclavate	16	6.82 × 10^5^
**BhEt9**	Fluffy Raised cottony	Grayish	79.6	Irregular	Grayish Black	Gray	Dark Brown	2–5	71.18	10	Obclavate	10	7.98 × 10^5^
**BeEt1**	Slightly Raised Fluffy	Whitish black	85.8	Irregular	Whitish Black	Black	Olivaceous brown	3–7	88.24	17.65	Slightly curved	8	8.85 × 10^5^
**BeEt2**	Flattened	Olivaceous brown	88.1	Regular	Brownish	Brown	Dark brown	2–5	74.71	11.76	Slightly curved	13	9.65 × 10^5^
**BeEt3**	Slightly Raised Fluffy	Blackish brown	90.0	Regular	Grayish Black	Brown	Olivaceous brown	3–9	104.71	15.29	Slightly curved	10	9.53 × 10^5^
**BeEt4**	Slightly Raised Fluffy	Olivaceous brown	89.2	Regular	Brownish	Brown	Dark brown	3–8	102.35	17.06	Obclavate	8	9.44 × 10^5^
**BeEt5**	Slightly Raised Fluffy	Olivaceous brown	64.6	Irregular	Whitish black	White	Olivaceous brown	4–10	122.35	15.29	Slightly curved	10	3.23 × 10^5^
**BeEt6**	Flattened	Grayish	90.0	Regular	Grayish Black	Gray	Brown	3–8	105.29	10	Slightly curved	10	9.88 × 10^5^
**BeEt7**	Flattened	Grayish	90.0	Regular	Grayish Black	Gray	Dark brown	3–11	116.47	16.47	Slightly curved	12	9.55 × 10^5^
**BeEt8**	Slightly Raised Fluffy	Olivaceous brown	87.1	Irregular	Whitish black	Gray	Olivaceous brown	3–7	120.94	20.06	Slightly curved	9	8.95 × 10^5^
**BeEt9**	Slightly Raised Fluffy	Grayish	90.0	Regular	Brownish	Gray	Olivaceous brown	3–10	117.65	15.88	Slightly curved	8	9.65 × 10^5^
**KhEt1**	Flattened	Grayish	90.0	Regular	Grayish Black	Gray	Brown	3–12	120.59	21.76	Slightly curved	12	9.54 × 10^5^
**KhEt2**	Fluffy Raised Cottony	Olivaceous brown	90.0	Regular	Brownish	White	Dark Brown	2–9	130.59	24.12	Slightly curved	18	9.48 × 10^5^
**KhEt3**	Flattened	Grayish	90.0	Regular	Grayish Black	Gray	Dark Brown	3–9	112.35	18.24	Slightly curved	9	9.68 × 10^5^
**KhEt4**	Fluffy Raised Cottony	Grayish	86.4	Regular	Grayish Black	Gray	Dark Brown	3–10	67.06	12.94	Slightly curved	22	8.86 × 10^5^
**KhEt5**	Slightly Raised Fluffy	Olivaceous brown	75.5	Irregular	Brownish	Brown	Brown	3–12	64.71	12.35	Slightly curved	26	7.12 × 10^5^
**KhEt6**	Fluffy Raised Cottony	Whitish black	80.6	Irregular	Whitish black	Black	Brown	3–10	69.41	11.76	Slightly curved	20	7.85 × 10^5^
**KhEt7**	Flattened	Olivaceous brown	80.9	Regular	Whitish black	Gray	Dark Brown	3–7	112.94	15.29	Slightly curved	10	7.89 × 10^5^
**KhEt8**	Flattened	Olivaceous brown	90.0	Regular	Brownish	Brown	Olivaceous brown	3–10	121.76	20	Slightly curved	16	9.78 × 10^5^
**KhEt9**	Flattened	Grayish	90.0	Regular	Grayish black	Gray	Brown	2–6	81.18	25.29	Slightly curved	12	9.75 × 10^5^
**KaEt1**	Slightly Raised Fluffy	Olivaceous brown	80.8	Regular	Brownish	White	Brown	2–5	99.41	21.76	Slightly curved	8	8.45 × 10^5^
**KaEt2**	Slightly Raised Fluffy	Whitish black	65.5	Irregular	Whitish black	Black	Olivaceous brown	3–8	116.47	20.59	Slightly curved	10	3.12 × 10^5^
**KaEt3**	Fluffy Raised Cottony	Olivaceous brown	85.6	Regular	Brownish	White	Brown	3–7	102.94	10.59	Slightly curved	8	8.92 × 10^5^
**KaEt4**	Fluffy Raised Cottony	Grayish	90.0	Regular	Grayish Black	Gray	Brown	3–12	65.29	11.76	Obclavate	25	9.6 × 10^5^
**KaEt5**	Fluffy Raised Cottony	Whitish black	64.7	Regular	Whitish black	White	Brown	3–7	77.06	20.59	Obclavate	20	3.08 × 10^5^
**KaEt6**	Fluffy Raised Cottony	Grayish	90.0	Regular	Grayish Black	Gray	Dark Brown	3–10	59.41	14.12	Obclavate	24	9.72 × 10^5^
**KaEt7**	Fluffy Raised Cottony	Grayish	86.4	Regular	Grayish Black	Gray	Olivaceous brown	2–8	77.65	10	Obclavate	13	9.42 × 10^5^
**KaEt8**	Fluffy Raised Cottony	Olivaceous brown	90.0	Regular	Whitish black	Gray	Olivaceous brown	3–6	100	18.82	Obclavate	15	9.66 × 10^5^
**KaEt9**	Slightly Raised Fluffy	Olivaceous brown	86.5	Regular	Brownish	Brown	Olivaceous brown	2–7	115.29	17.06	Obclavate	12	8.96 × 10^5^
**SaEt1**	Fluffy Raised Cottony	Grayish	90.0	Regular	Grayish Black	Gray	Dark Brown	3–8	55.29	15.29	Obclavate	24	9.88 × 10^5^
**SaEt2**	Fluffy Raised Cottony	Grayish	90.0	Regular	Grayish Black	Gray	Dark Brown	3–12	87.65	13.53	Obclavate	20	9.84 × 10^5^
**SaEt3**	Flattened	Olivaceous brown	80.1	Regular	Brownish	Brown	Brown	3–12	59.41	16.47	Slightly curved	16	8.75 × 10^5^
**SaEt4**	Slightly Raised Fluffy	Blackish brown	89.9	Regular	Grayish Black	Brown	Brown	3–10	144.12	25.88	Slightly curved	13	9.35 × 10^5^
**SaEt5**	Flattened	Blackish brown	90.0	Regular	Grayish Black	Brown	Dark Brown	3–8	83.53	19.41	Obclavate	16	9.76 × 10^5^
**SaEt6**	Fluffy Raised Cottony	Grayish	90.0	Regular	Grayish Black	Gray	Olivaceous brown	2–9	102.35	11.18	Slightly curved	22	9.32 × 10^5^
**SaEt7**	Fluffy Raised Cottony	Grayish	90.0	Regular	Grayish Black	Gray	Dark Brown	3–10	110.59	18.82	Slightly curved	20	9.47 × 10^5^
**SaEt8**	Flattened	Grayish	89.8	Regular	Grayish Black	Gray	Brown	4–9	81.18	15.88	Slightly curved	26	9.12 × 10^5^
**SaEt9**	Flattened	Olivaceous brown	86.8	Regular	Brownish	Brown	Brown	3–10	52.94	11.18	Slightly curved	15	9.44 × 10^5^

## Data Availability

The original data were available from the corresponding author upon an appropriate request.

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
