# Peer review of "Molecular and Morphological Characterization of Exserohilum turcicum (Passerini) Leonard and Suggs Causing Northern Corn Leaf Blight of Maize in Bihar"

_bioengineering, 2022, doi:10.3390/bioengineering9080403_

Round 1
Reviewer 1 Report
The paper submitted by Anwer et al. is focused on the determination of molecular and morphological features of E. turcicum. the paper is interesting and well written. It should be published in Bioengineering after addressing some comments listed below:
1) the quality of Fig. 1 nad 3 should be increased - in the present form it is difficult for the reader to observed differences between samples based on presented photos
2) font in Table 1 should be unified with the whole manuscript
3) Figure 4 - please show based on PCA analysis which features of analysed material were the most important for the observed clustering - appropriate figure would be useful
Author Response
Revered Reviewer,
We would like to thank reviewer for valuable suggestions, insightful comments and inputs have considerably improved the manuscript. Last, but not the least, we thank you once again for considering our manuscript for publication in your esteemed journal. Please find our point-by-point response (in red) to reviewer comments along with this. Further, we have made the necessary changes in the manuscript (in track changes version of revised manuscript).
We are looking forward to hearing from you.
Thanks and regards
Md. Arshad Anwer

Reviewer 2 Report
Dear authors
Happy day.
The paper can be improved significantly after some efforts.
- More references are needed in this paper particularly in the Material and methods parts.
- You did not show information about the primers you use and which part they amplify. Did they amplify a relatively conserved region?
- Why you use another type of tree. Kindly use the same type and show if there is a differences between the tree which based on the morphological data and that one which based on the DNA sequences.
- why the backgrounds of the sample are different as well as the magnifications or there are some species show smaller spores!
- Kindly, put a single image that represent the best morphology of a single group for each together to be a reference for differentiation for any one need to use them.
- Kindly, search the other studies to prove or disprove that the differnt fungus have different characters, for example "against the different antimicrobial agents".
- Kindly, show the DNA sequences or part of them as alignments to prove the differences between the different isolates.
- I suggest that you should show whether you get similar or different DNA sequences. I suggest to build another tree based on the DNA sequences you get and compare it with that you get from the morphological studies. This is the main point in the paper.
- If you get a conserved sequence you should clarify that.
- You did not compare your work with a single previous study! Did you think that you are the first? if not so kindly put in the result and the discussion part a comparison between your work and the other studies?
Finley, all this suggestions should improve your work significantly, they all did not any experimental study but you can use the data you have to put your paper in a more better style.
Good luck.
With my pleasure
Amro Amara

Author Response
Revered Reviewer,
We would like to thank reviewer for valuable suggestions, insightful comments and inputs have considerably improved the manuscript. Last, but not the least, we thank you once again for considering our manuscript for publication in your esteemed journal. Please find our point-by-point response (in red) to reviewer comments along with this and a supplementary figure 1. Further, we have made the necessary changes in the manuscript (in track changes version of revised manuscript).
We are looking forward to hearing from you.
Thanks and regards
Md. Arshad Anwer

Round 2
Reviewer 2 Report
Dear all
Happy day
Many thanks for considering all the points in my comment.
The paper is fine now,
With my pleasure
Amro Amara